# Causal Inference via Kernel Deviance Measures

**Jovana Mitrovic**∗    **Dino Sejdinovic**    **Yee Whye Teh**∗
Department of Statistics, University of Oxford
[mitrovic, dino.sejdinovic, y.w.teh]@stats.ox.ac.uk

## Abstract

Discovering the causal structure among a set of variables is a fundamental problem in many areas of science. In this paper, we propose Kernel Conditional Deviance for Causal Inference (KCDC) a fully nonparametric causal discovery method based on purely observational data. From a novel interpretation of the notion of asymmetry between cause and effect, we derive a corresponding asymmetry measure using the framework of reproducing kernel Hilbert spaces. Based on this, we propose three decision rules for causal discovery. We demonstrate the wide applicability and robustness of our method across a range of diverse synthetic datasets. Furthermore, we test our method on real-world time series data and the real-world benchmark dataset Tübingen Cause-Effect Pairs where we outperform state-of-the-art approaches.

## 1 Introduction

In many areas of science, we strive to answer questions that are fundamentally causal in nature. For example, in medicine one is often interested in the genetic drivers of diseases, while in commerce one might want to identify the motives behind customers' purchasing behaviour. Furthermore, it is of the utmost importance to thoroughly understand the underlying causal structure of the data-generating process if we are to predict, with reasonable accuracy, the consequences of interventions or answer counterfactual questions about what would have happened had we acted differently. While most machine learning methods excel at prediction tasks by successfully inferring statistical dependencies, there are still many open questions when it comes to uncovering the causal dependencies between the variables driving the underlying data-generating process. Given the growing interest in using data to guide decisions in areas where interventional and counterfactual questions abound, causal discovery methods have attracted considerable research interest [9, 25, 13, 16].

While causal inference is preferably performed on data coming from randomized control experiments, often this kind of data is not available due to a combination of ethical, technical and financial considerations. These real-world limitations have motivated research into inferring causal relationships from purely observational data. While methods that attempt to recover the causal structure by analyzing conditional independencies present in the data [20, 23] are mathematically well-founded, they are not robust to the choice of conditional independence testing methodology. Another group of methods [9, 25, 14] postulates that there is some inherent asymmetry between cause and effect and proposes different asymmetry measures that form the basis for causal discovery. In order to facilitate causal inference, these approaches typically assume a particular functional form for the interaction between the variables and a particular noise structure which limits their applicability. We aim our contribution to be a step towards a method that can deal with highly complex data-generating processes, relies only observational data and whose inference can easily be extended without the need to develop novel, specifically tailored algorithms for each new model class.

In this work, we develop a fully nonparametric causal inference method to automatically discover causal relationships from purely observational data. In particular, our proposed method does not

---

∗Now at DeepMind, UK.

require any *a priori* assumptions on the functional form of the interaction between the variables or the noise structure. Furthermore, we propose a novel interpretation of the notion of asymmetry between cause and effect [4]. Before we introduce our proposed interpretation, we motivate it with the following example. Let $y = x^3 + x + \epsilon$ with $\epsilon \sim \mathcal{N}(0, 1)$ where we consider the correct causal direction to be $x \rightarrow y$. Figure 1 visualizes the conditional distributions $p(y|x)$ and $p(x|y)$ for different values of $x$ and $y$, respectively.

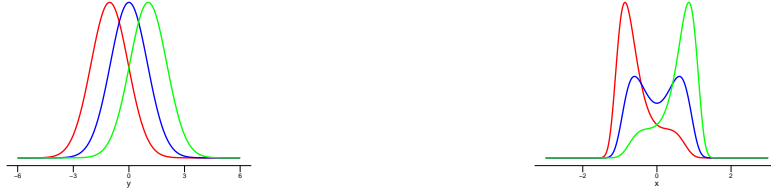

Figure 1: Conditional distributions $p(y|x)$ and $p(x|y)$ for different values of the conditioning variables $x$ and $y$, respectively. These represent the causal and anticausal direction, respectively.

Note that the conditional distributions in the anticausal direction exhibit a larger structural variability across different values of the conditioning variable than the conditional distributions in the causal direction. It is important to note here that structural variability does not only refer to variability in the scale and location parameters, but should be understood more broadly as variability in the "parametric" form, e.g. differences in the number of modes and in higher order moments. If one thinks of conditional distributions as programs generating $y$ from $x$ and vice versa, we see that in the causal direction the structure of the program remains unchanged although different input arguments are provided. In the anticausal direction, the program requires structural modification across different values of the input in order to account for the differing behaviour of the conditional densities.

Motivated by the above observation, we popose a novel interpretation of the notion of asymmetry between cause and effect in terms of the shortest description length, i.e. Kolmogorov complexity [8], of the data-generating process. Whereas previous work [11, 10, 4, 2] quantifies the asymmetry in terms of the Kolmogorov complexity of the factorization of the joint distribution, we propose to interpret the asymmetry based on the Kolmogorov complexity of the conditional distribution. Specifically, we propose that this asymmetry is realized by *the Kolmogorov complexity of the mechanism in the causal direction being independent of the input value of the cause*. On the other hand, in the anticausal direction, there will be a dependence between the shortest description length of the mechanism and the particular input value of the effect. This (in)dependence can be measured by looking at the variability of Kolmogorov complexities of the mechanism for particular of the input. Unfortunately, as computing the Kolmogorov complexity is an intractable problem, we resort to conditional distributions as approximations of the corresponding programs. Thus, we can infer the causal direction by comparing the description length variability of conditional distributions across different values of the conditioning variable with the causal direction being the less variable. In particular, we propose three decision rules for causal inference based on this criterion. For measuring this variability, we use the framework of reproducing kernel Hilbert spaces (RKHS). This allows us to represent conditional distributions in a compact, yet expressive way and efficiently capture their many nuanced aspects thus enabling more accurate causal inference. In particular, by way of the kernel trick, we can efficiently compute the variability of infinite-dimensional objects using finite-dimensional quantities that can be easily estimated from data. Using the RKHS framework makes our method readily applicable also in situations when trying to infer the causal direction between pairs of random variables taking values in structured or non-Euclidean domains on which a kernel can be defined.

The main contributions of this paper are:

- an interpretation of the notion of asymmetry between cause and effect in terms of the independence of the description length of the mechanism on the value of the cause,

- an approximation to the intractable description length in terms of conditional distributions,

- a flexible asymmetry measure based on RKHS embeddings of conditional distributions,

- a fully nonparametric method for causal inference that does not impose *a priori* any assumptions on the functional relationship between the variables or the noise structure.

## 2 Related Work

Most approaches to causal inference from purely observational data can be grouped into three categories. Constraint-based methods assume that the true causal structure can be represented with a directed acyclic graph (DAG) $G$ which they try to infer by analyzing conditional independencies present in the observational data distribution $P$. Under some technical assumptions [17], these methods can determine $G$ only up to its Markov equivalence class[2] which usually contains DAGs that can be structurally very diverse and still have many unoriented edges. Examples of this methodology include [23, 26] which rely on kernel-based conditional independence criteria and the PC algorithm [20] builds a graph skeleton by successively removing unnecessary connections between the variables and then orienting the remaining edges if possible. Although mathematically well-founded, the performance of these methods is highly dependent on the utilized conditional independence methodology, whose performance usually depends on the amount of available data. Furthermore, these methods are not robust as small errors in building the graph skeleton (e.g. a missing independence relation) can lead to significant errors in the inferred Markov equivalence class. As conditional independence tests require at least three variables, they are not applicable in the two variable case.

Score-based methods search the space of all DAGs of a certain size by scoring their fit to the observed data using a predefined score function. An example of this approach is Greedy Equivalent Search [3] which combines greedy search with the Bayesian information criterion. As the search space grows super-exponentially with the number of variables, these methods quickly become computationally intractable. An answer to this shortcoming are hybrid methods which use constraint-based approaches to decrease the search space that can then be effectively explored with score-based methods, e.g. [24]. DAGs have also been represented using generative neural networks and scored according to how well the generated data matches the observed data, e.g. [7]. A major shortcoming of this hybrid methodology is that there exists no principled way of choosing problem-specific combinations of scoring functions and search strategies which is a significant problem as different search strategies in combination with different scoring rules can potentially lead to very different results.

The third category of methods assumes that there exists some inherent asymmetry between cause and effect. So-called functional causal models or structural equation models assume a particular functional form for the causal interactions between the variables and a particular noise structure. In these models, each variable is a deterministic function of its causes and some independent noise, with all noise variables assumed to be jointly independent. Examples of this methodology assume linearity and additive non-Gaussian noise [19], nonlinear additive noise [9, 14] and invertible interactions between the covariates and the noise [25]. In order to perform causal discovery in these models, the special structural assumptions placed on the interaction between the covariates and on the noise are of crucial importance, thus limiting their applicability. A second strand of research interprets the asymmetry between cause and effect through an information-theoretic lens by examining the complexity of the factorization of the joint distribution [11]. [10] argue that if $X$ causes $Y$, then the factorization in the causal direction, i.e. $p(X,Y) = p(Y|X)p(X)$, should have a shorter description in terms of the Kolmogorov complexity than the factorization in the anticausal direction, i.e. $p(X,Y) = p(X|Y)p(Y)$. In [4], instead of computing the intractable Kolmogorov complexity, the correlation between the input and the conditional distribution is measured, whereas [2] use the minimum description length principle. The approach of [22] measures the complexity of conditional distributions by RKHS seminorms computed on the logarithms of their densities.

Lastly, causal discovery has also been framed as a learning problem. RCC [13] uses feature representations of the data based on RKHS embeddings of the joint and marginal distributions within a random forest classifier. In [5], the feature representation includes quantities describing the joint, marginal and conditional distributions. In particular, the conditional distributions are represented with conditional entropy, mutual information and a quantification of their variability in terms of the spread of the entropy, variance and skewness for different values of the conditioning variable. This differs from our approach where we base our causal inference method on a novel interpretation of the asymmetry between cause and effect, and based on this derive three decision rules with one of these decision rules relying on classifying feature representations. In particular, we consider feature representations based only on conditional distributions which we argue to be more discriminative for inferring the causal direction.

# 3 Kernel Conditional Deviance for Causal Inference

## 3.1 Background

Let $(\mathcal{X}, \mathcal{B}_\mathcal{X})$ and $(\mathcal{Y}, \mathcal{B}_\mathcal{Y})$ be measurable spaces with $\mathcal{B}_\mathcal{X}$ and $\mathcal{B}_\mathcal{Y}$ the associated Borel $\sigma$-algebras. Denote by $(\mathcal{H}_\mathcal{X}, k)$ and $(\mathcal{H}_\mathcal{Y}, l)$ the RKHSs of functions defined on $\mathcal{X}$ and $\mathcal{Y}$, respectively, and their corresponding kernels. Given a probability distribution $p$ on $\mathcal{X}$, the *mean embedding* $\mu_p$[3] [18] is a representation of $p$ in $\mathcal{H}_\mathcal{X}$ given by $\mu_p = \mathbb{E}_p[k(\cdot, X)]$ with $X \sim p$. It can be unbiasedly estimated by $\hat{\mu}_p = \frac{1}{n} \sum_{i=1}^n k(\cdot, x_i)$ with $\{x_i\}_{i=1}^n \overset{iid}{\sim} p$. Furthermore, if $k$ is a characteristic kernel [18], then this representation yields a metric on probability measures, i.e. $\|\mu_p - \mu_q\|_{\mathcal{H}_k} = 0 \Leftrightarrow p = q$. A conditional distribution $p(X|Y = y)$ can be encoded using the *conditional mean embedding* $\mu_{X|Y=y}$ [18] which is an element of $\mathcal{H}_\mathcal{X}$ that satisfies $\mathbb{E}[h(X)|Y = y] = \langle h, \mu_{X|Y=y} \rangle_{\mathcal{H}_\mathcal{X}} \quad \forall h \in \mathcal{H}_\mathcal{X}$. Using the equivalence between conditional mean embeddings and vector-valued regressors [12], we can estimate $\mu_{X|Y=y}$ from a sample $\{(x_i, y_i)\}_{i=1}^n \overset{iid}{\sim} p(x, y)$ as $\hat{\mu}_{X|Y=y} = \sum_{i=1}^n \alpha_i(y) k(\cdot, x_i)$ with regularization parameter $\lambda$ and identity matrix $\mathbf{I}$, $\alpha(y) = (\mathbf{L} + n\lambda\mathbf{I})^{-1}\mathbf{l}_y$, $\mathbf{L} = [l(y_i, y_j)]_{i,j=1}^n$, $\mathbf{l}_y = [l(y_1, y), \ldots, l(y_n, y)]^T$, $\alpha(\cdot) = [\alpha_1(\cdot), \ldots, \alpha_n(\cdot)]^T$. For a more detailed discussion, see [18].

## 3.2 Method

For simplicity, we restrict our attention to the two variable problem of causal discovery, i.e. distinguishing between cause and effect. Possible extensions to the multivariable setting are discussed in Section 5. Following the usual approach in the literature, we derive our method under the assumption of causal sufficiency of the data. In particular, we ignore the potential existence of confounders, i.e. all causal conclusions should be understood with respect to the set of observed variables. Nevertheless, in Section 4, we see that our method performs well also in settings where the noise has positive mean which can be interpreted as accounting for potential confounders.

Given observations $\{(x_i, y_i)\}_{i=1}^n$ of a pair of random variables $(X, Y)$, our goal is to infer the causal direction, i.e. decide whether $X$ causes $Y$ (i.e. $X \to Y$) or $Y$ causes $X$ (i.e. $Y \to X$). To this end, we develop a fully nonparametric causal discovery method that relies only on observational data. In particular, our method does not *a priori* postulate a particular functional model for the interactions between the variables or a particular noise structure. Our approach, Kernel Conditional Deviance for Causal Inference (KCDC), is based on the assumption that there exists an asymmetry between cause and effect that is inherent in the data-generating process. While there are many interpretations of how this asymmetry might be realized, two of the more prominent ideas phrase it in terms of the independence of cause and mechanism [4] and in terms of the complexity of the factorization of the joint distribution [11, 10].

Motivated by these two ideas, we propose a novel interpretation of the notion of asymmetry between cause and effect. First, we take an information-theoretic approach to reasoning about the complexity of distributions similar to [11, 10]. In particular, we reason about it in terms of algorithmic complexity, i.e. Kolmogorov complexity [8] which is the description length of the shortest program that implements the sampling process of the distribution. For a distribution $p(Y)$, the Kolmogorov complexity is

$$K(p(Y)) = \min_s \{|s| : |U(s, y, q) - p(y)| \le q \ \forall y\}$$

with $q$ a precision parameter, $U(\cdot)$ extracting the output of applying program $s$ onto a realization of the random variable $Y$ denoted by $y$. Analogously, for a conditional distribution $p(Y|X)$, the Kolmogorov complexity is

$$K(p(Y|X)) = \min_s \{|s| : |U(s, y, x, q) - p(y|x)| \le q \ \forall x, y\}.$$

Assuming $X \to Y$, the asymmetry notion specified in terms of factorization complexity can be expressed as

$$K(p(X)) + K(p(Y|X)) \le K(p(Y)) + K(p(X|Y))$$

which holds up to an additive constant [21]. Further, the independence of cause and mechanism can be interpreted as algorithmic independence [10], i.e. knowing the distribution of the cause $p(X)$ does not enable a shorter description of the mechanism $p(Y|X)$.

Based on this, we argue that not only knowing the distribution of the cause does not enable a shorter description of the mechanism, but also knowing any particular value of the cause does not provide any information that can be used to construct a shorter description of the mechanism. To formalize this, we introduce the notation

$$K(p(Y|X = x)) = \min_{s}\{|s| : |U(s, y, x, q) - p(y|X = x)| \leq q \ \forall y\}$$

to be the Kolmogorov complexity of the conditional distribution $p(Y|X)$ when the conditioning variable takes on the value $X = x$. From our argument above, we see that in the causal direction the Kolmogorov complexity of $p(Y|X = x)$ is independent of the particular value $x$ of the cause $X$, i.e.

$$K(p(Y|X = x_i)) = K(p(Y|X = x_j)) \quad \forall i, j.$$

On the other hand, this will not hold in the anticausal direction as the input and mechanism are not algorithmically independent in that direction, i.e.

$$K(p(X|Y = y_i)) \neq K(p(X|Y = y_j)) \quad \forall i \neq j.$$

This motivates our interpretation of the notion of asymmetry between cause and effect which is summarized as follows.

**Postulate.** *(Minimal description length independence)*
*If $X \to Y$, the minimal description length of the mechanism mapping $X$ to $Y$ is independent of the value of $X$, whereas the minimal description length of the mechanism mapping $Y$ to $X$ is dependent on the value of $Y$.*

Building on this, we can infer the causal direction by comparing how much the description length of the minimal description length program implementing the mechanism varies across different values of its input arguments. In particular, in the causal direction, we expect to see less variability than in the anticausal direction. As computing the Kolmogorov complexity is an intractable problem, we use th norm of RKHS embeddings of the corresponding conditional distributions as a proxy for it. Thus, we recast causal inference in terms of comparing the variability in RKHS norm of embeddings of sets of conditional distributions indexed by values of the conditioning variable. In order to perform causal inference, we use the framework of reproducing kernel Hilbert spaces. This allows us to construct highly expressive, yet compact approximations of the potentially highly-complex programs and circumvent the challenges of density estimation when trying to represent conditional distributions. Furthermore, using the RKHS framework allows us to efficiently capture the many nuanced aspects of distributions thus enabling more accurate causal inference. For example, using non-linear kernels allows us to capture more comprehensive distributional properties including higher order moments. Furthermore, using the RKHS framework makes our method readily applicable also in situations when trying to infer the causal direction between two random vectors (treated as single variables) or pairs of other types of random variables taking values in structured or non-Euclidean domains on which a kernel can be defined. Examples of such types of data include discrete data, genetic data, phylogenetic trees, strings, graphs and other structured data [6].

We represent conditional distributions in the RKHS using conditional mean embeddings [18]. In particular, given observations $\{(x_i, y_i)\}_{i=1}^{n}$ of a pair of random variables $(X, Y)$, we construct the embeddings of the two sets of conditional distributions, $\{p(X|Y = y_i)\}_{i=1}^{n}$ and $\{p(Y|X = x_i)\}_{i=1}^{n}$. Furthermore, if we choose a characteristic kernel [18], the conditional mean embeddings of two distinct distributions will not overlap. Next, we compute the variability in RKHS norm of a set of conditional mean embeddings as the deviance of the RKHS norms of that set. Thus, using the KCDC measure $S_{X \to Y}$ with

$$S_{X \to Y} = \frac{1}{n}\sum_{i=1}^{n}\left(\left\|\mu_{Y|X=x_i}\right\|_{\mathcal{H}_{\mathcal{Y}}} - \frac{1}{n}\sum_{j=1}^{n}\left\|\mu_{Y|X=x_j}\right\|_{\mathcal{H}_{\mathcal{Y}}}\right)^{2}, \tag{1}$$

we compute the deviance in RKHS norm of the set $\{p(Y|X = x_i)\}_{i=1}^{n}$. For $\{p(X|Y = y_i)\}_{i=1}^{n}$, we analogously compute the KCDC measure $S_{Y \to X}$.

Based on our proposed interpretation of the notion of asymmetry between cause and effect, we can determine the causal direction between $X$ and $Y$. Furthermore, we derive a confidence measure $\mathcal{T}^{\mathcal{KCDC}}$ for the inferred causal direction as

$$\mathcal{T}^{\mathcal{KCDC}} = \frac{|S_{X \to Y} - S_{Y \to X}|}{\min(S_{X \to Y}, S_{Y \to X})}.$$

To determine the causal direction, we propose three decision rules. In particular, we can determine the causal direction by directly comparing the KCDC measures for the two directions, i.e.

$$D_1(X, Y) = \begin{cases} X \to Y, & \text{if} \quad S_{X \to Y} < S_{Y \to X} \\ Y \to X, & \text{if} \quad S_{X \to Y} > S_{Y \to X} \end{cases}$$

but leave the causal direction undetermined if the KCDC measures are too close in value to determine the causal direction, i.e. $\mathcal{T}^{\mathcal{KCDC}} < \delta$ with $\delta$ some fixed decision threshold. This situation might come about due to numerical errors or non-identifiability. We can also determine the causal direction based on majority voting of an ensemble constructed using different model hyperparameters, i.e.

$$D_2(X, Y) = \texttt{Majority}(\{D_1^{H_j}(X, Y)\}_j)$$

where the dependence on the model hyperparameters $H_j$ has been made explicit. Lastly, the KCDC measures can also be used for constructing feature representations of the data which can then be used within a classification method. In particular, we can infer the causal relationship between $X$ and $Y$ using

$$D_3(X, Y) = \texttt{Classifier}(\{S_{X \to Y}^{H_j}, S_{Y \to X}^{H_j}\}_j)$$

where Classifier is a classification algorithm that classifies $X \to Y$ against $Y \to X$. For training the classifier, we generate synthetic data, e.g. as in [13]. Algorithms summarizing our causal inference methodology are given in the supplementary material.

**Identifiability.** In order to ensure the identifiability of the model in KCDC, we need to ensure that we apply the same kernel to the response variable when computing the KCDC measures $S_{X \to Y}$ and $S_{Y \to X}$. Specifically, we fix a kernel $k_r$ with some fixed hyperparameters and apply it both to $Y$ when computing $S_{X \to Y}$ and to $X$ when computing $S_{Y \to X}$. This ensures that we are measuring the variability in both directions in the same space, thus making them comparable. Furthermore, we need to also apply the same fixed kernel $k_{in}$ (with some fixed hyperparameters) to the input variable, i.e. to $X$ for $S_{X \to Y}$ and to $Y$ for $S_{Y \to X}$. This ensures that the set of possible functional dependencies between input and response is the same in both directions. In particular, due to the equivalence of conditional mean embeddings and vector-valued regressors [12], this is analogous to the requirement for constraining the assumed functional class in structural equation models in order to ensure identifiability [25]. Given the postulate of minimal description length independence which is the basis for causal discovery in KCDC, the only case when the causal direction will not be identifiable for KCDC is the situation where the description length of conditional distributions in both the causal and anticausal direction does not vary with the value of the cause and effect, respectively. This happens when in both directions the functional form of the mechanism can be described by one family of distributions for all its input arguments. One example of this is linear Gaussian dependence which is non-identifiable for most other causal discovery methods too. Another example is the case of independent variables which is usually not considered in the literature, but can be easily mitigated with an independence test. Note that using characteristic kernels eliminates any potential non-identifiability that might arise as a consequence of the non-injectivity of the embedding.

## 4 Experimental Results

### 4.1 Synthetic Data

In order to showcase the wide applicability and robustness of our proposed approach, we test it extensively on several synthetic datasets spanning a wide range of functional dependencies between cause and effect and different interaction patterns with different kinds of noise. Table 1 summarizes the different models used to generate synthetic data. In all of the below experiments, we sample 100 datasets of 100 observations each with $x \sim \mathcal{N}(0, 1)$ and test three different noise regimes – either $\epsilon \sim \mathcal{N}(0, 1)$, $\epsilon \sim \mathcal{U}(0, 1)$ or $\epsilon \sim \text{Exp}(1)^4$.

Table 1: Summary of the different functional models used for generating synthetic data.

| | Additive Noise | Multiplicative Noise | Complex Noise |
|---|---|---|---|
| (A) | $y = x^3 + x + \epsilon$ | $y = (x^3 + x)e^\epsilon$ | $y = (\log(x + 10) + x^2)^\epsilon$ |
| (B) | $y = \log(x + 10) + x^6 + \epsilon$ | $y = (\sin(10x) + e^{3x})e^\epsilon$ | $y = \log(x + 10) + x^{2|\epsilon|}$ |
| (C) | $y = \sin(10x) + e^{3x} + \epsilon$ | $y = (\log(x + 10) + x^6)e^\epsilon$ | $y = \log(x^7 + 5) + x^5$ |
| | | | $\quad - \sin(x^2|\epsilon|)$ |

We compare our approach to LiNGAM [19], IGCI [4], ANM [16] with Gaussian Process regression and HSIC test [18] on the residual and the post-nonlinear model (PNL) [25] with HSIC test. In all experiments, we apply the decision rule based on direct comparison for KCDC. We tested across different combinations of characteristic kernels (radial basis function (RBF), log and rational quadratic kernels[5]) which yielded fairly consistent performance. We report the results when using the log kernel on the input and the rational quadratic kernel on the response.

Table 1 summarizing the results is given in the supplementary material. LiNGAM performs badly across most of the settings which is to be expected given given its assumption of linear dependence. ANM performs very well under additive noise across the different noise settings, but displays bad performance under complex and, especially, multiplicative noise due to its assumption of additive noise. On the other hand, PNL does not perform well under additive noise which is probably due to overfitting. It performs slighly better under multiplicative noise, but does not surpass chance level in half the settings. Under complex noise, PNL, which assumes a invertible interaction between the covariates and noise, performs at or above chance level in almost all cases with very good performance under periodic noise. In additive noise settings, IGCI performs well for model (C) and under exponential noise across all models, while in the multiplicative and complex noise settings, it shows excellent performance. Our proposed method achieves perfect performance in all settings of additive and multiplicative noise across all noise regime. Under complex noise, it achieves perfect performance in all cases except under Gaussian and uniform noise for (A).

## 4.2 Tübingen Cause-Effect Pairs

Next, we discuss the performance of our method on real-world data. For this purpose, we test KCDC on the only widely used benchmark dataset Tübingen Cause-Effect Pairs (TCEP) [15]. This dataset is comprised of real-world cause-effect samples that are collected across very diverse subject areas with the true causal direction provided by human experts. Due to the heterogenous origins of the data pairs, many diverse functional dependencies are expected to be present in TCEP.

In order to show the flexibility and capacity of KCDC when dealing with many diverse functional dependencies simultaneously, we test it using both the direct comparison decision rule and the majority decision rule. We use TCEP version 1.0 which consists of 100 cause-effect pairs. Each pair is assigned a weight in order to account for potential sources of bias given that different pairs are sometimes selected from the same multivariable dataset. Following the wide-spread approach present in the literature of testing only on scalar-valued pairs, we remove the multivariate pairs 52, 53, 54, 55 and 71 from TCEP in order to ensure a fair comparison to previous work. Note that contrary to some methods in literature, this is not necessary for our approach. For the majority approach, we choose the best settings of the kernel hyperparameters as inferred from the synthetic experiments. The direct approach represents the single best performing hyperparameter configuration on TCEP.

From the summary of classification accuracies of KCDC and related methods in Table 2, we see that KCDC is competitive to the state-of-the-art methods even when only one setting of kernel hyperparameters is used, i.e. when the direct comparison decision rule is used. When we combine multiple kernel hyperparameters under the majority vote approach, we see that our method outperforms other methods by a significant margin. Note that the review [16] discusses additive noise models [9] and information-geometric causal inference [4]. In particular, an extensive experimental evaluation of these methods across a wide range of hyperparameter settings is performed. In the fourth row of

Table 2, we report the most favourable outcome across both types of methods of their large-scale experimental analysis. For testing RCC on TCEP v1.0, we use the code provided in [13].

Table 2: Classification Accuracy On TCEP

| ANM | PNL | RCC | Best from [16] | CGNN [7] | KCDC-$D_1$ | KCDC-$D_2$ |
|---|---|---|---|---|---|---|
| 59.5% | 66.2% | 64.67% | ≈ 74% | 74.4% | 72.87% | **78.71%** |

### 4.3 Inferring the Arrow of Time

In addition to the many real-world pairs above, we also test our method at inferring the direction of time on causal time series. Given a time series $\{X_i\}_{i=1}^N$, the task is to infer if $X_i \to X_{i+1}$ or $X_i \leftarrow X_{i+1}$. We use a dataset containing quarterly growth rates of the real gross domestic product (GDP) of the UK, Canada and USA from 1980 to 2011 as in [1]. The resulting multivariate time series has length 124 and dimension three. According to the above selection of hyperparameters on the synthetic datasets, we chose a wide range of hyperparameters to test KCDC on. In particular, both on the response and input we used either a log kernel $k(x, x') = -\log(\|x - x'\|^q + 1)$ with $q$ in $[2, 3, 4]$ or an RBF kernel with bandwidth $[1, 1.5, 2]$ times the median heuristic. Across all of these hyperparameters, KCDC correctly identifies the causal direction with the confidence measure $\mathcal{T}^{KCDC}$ measuring the absolute relative difference between the KCDC measures varying between 2.45 and 44565.6. We compare our approach to methods readily applicable to causal infenrence on multivariable time series. In particular, LiNGAM does not identify the correct direction. On the other hand, the method developed in [1] that models the data as an autoregressive moving average model with non-Gaussian noise correctly identifies the causal direction.

## 5 Extensions to the Multivariable Case

While we present and discuss our method for the case of pairs of variables, it can be extended to the setting of more than two variables. Assuming we have $d$ variables with $d \geq 2$, i.e. $\mathbb{X} = \{X_1, \ldots, X_d\}$, we can apply KCDC to every pair of variables $\{X_i, X_j\} \subseteq \mathbb{X}$ with $i \neq j$ while conditioning on all of the remaining variables in $\mathbb{X} \setminus \{X_i, X_j\}$. This corresponds to inferring the causal relationship between $X_i$ and $X_j$ while accounting for the confounding effect of all the remaining variables.

Another way of dealing with the multivariable setting is to use KCDC in conjunction with, for example, the PC algorithm [20]. In particular, one would first apply the PC algorithm to the data. The resulting DAG skeleton containing potentially many unoriented edges can then be processed with KCDC. In particular, our method can be applied sequentially to every pair of variables that is connected with an unoriented edge while conditioning on the remaining variables in the DAG.

Yet another approach to the multivariable case is to use KCDC measures as features in a multiclass classification problem for $d$-dimensional distributions. However, as noted in [13], this approach quickly becomes rather cumbersome as the number of labels grows super-exponentially in the number of variables due to the rapid increase of the number of DAGs that can be constructed from $d$ variables.

## 6 Conclusion

In this paper, we proposed a fully nonparametric causal inference method that uses purely observational data and does not postulate *a priori* assumptions on the functional relationship between the variables or the noise structure. We proposed a novel interpretation of the notion of asymmetry between cause and effect in terms of the variability, across different values of the input, of the minimal description length of programs implementing the data-generating process of conditional distributions. In order to quantify the description length variability, we proposed a flexible measure in terms of the within-set deviance of the RKHS norms of conditional mean embeddings and presented three decision rules for causal inference based on direct comparison, ensembling and classification, respectively. We extensively tested our proposed method across a wide range of diverse synthetic datasets showcasing its wide applicability and robustness. Furthermore, we tested our method on real-world time series data and the real-world benchmark dataset Tübingen Cause-Effect Pairs where we outperformed existing state-of-the-art methods by a significant margin.

**Acknowledgments**

JM acknowledges the financial support of The Clarendon Fund of the University of Oxford. DS's and YWT's research leading to these results has received funding from the European Research Council under the European Union's Seventh Framework Programme (FP7/2007–2013) ERC grant agreement no. 617071.

## Footnotes

[2]All DAGs that encode the same set of conditional independence relations constitute a Markov equivalence class.

[3] $\mu_p$ and $\mu_X$ will be used interchangeably if it does not lead to confusion.

[4]Note that the exponential noise has positive mean which can be interpreted as accounting for confounders.

[5]The RBF, log and rational quadratic kernel are defined as $k(x, x') = \exp\left(-\|x - x'\|^2/(2\sigma^2)\right)$ with bandwidth $\sigma$, $k(x, x') = -\log(\|x - x'\|^2 + 1)$ and $k(x, x') = 1 - \|x - x'\|^2/(\|x - x'\|^2 + 1)$, respectively.

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
