[Supplementary Material · kcdc_neurips_supp.pdf]

# Supplementary Material for
# Causal Inference via Kernel Deviance Measures

## 1 Algorithmic description of the KCDC Method

---
**Algorithm 1** KCDC Algorithm

---
**Input:** Realizations $\{(x_i, y_i)\}_{i=1}^n$ of $(X, Y)$
**Output:** Causal direction ($X \to Y$ vs. $Y \to X$)

Determine causal direction using one of the following

    (A)  Compute $S_{X \to Y}$ and $S_{Y \to X}$ with Algorithm 2
          Perform direct comparison with decision rule $D_1$

    (B)  Compute $S_{X \to Y}$ and $S_{Y \to X}$ with Algorithm 2 for model hyperparameters $\{H_j\}_j$
          Perform majority voting with decision rule $D_2$

    (C)  Compute $S_{X \to Y}$ and $S_{Y \to X}$ with Algorithm 2 for model hyperparameters $\{H_j\}_j$
          Build data representation with $\{S_{X \to Y}^{H_j}, S_{Y \to X}^{H_j}\}_j$
          Train `Classifier` using synthetic data and use decision rule $D_3$

---

---
**Algorithm 2** Compute KCDC Measures

---
**Input:** Realizations $\{(x_i, y_i)\}_{i=1}^n$ of $(X, Y)$, model hyperparameters $H_j$
**Output:** KCDC measures $S_{X \to Y}$ and $S_{Y \to X}$

**for** $i = 1, \ldots, n$ **do**
    Embed $\{p(Y|X = x_i)\}_{i=1}^n$ and $\{p(X|Y = y_i)\}_{i=1}^n$ using model hyperparameters $H_j$
**end for**
Compute $S_{X \to Y}$ and $S_{Y \to X}$

---

# 2 Experimental results for synthetic datasets

Table 3: Classification Accuracies over 100 synthetic datasets across different functional interactions between the response and the covariates (A)-(C), different interaction patterns between the noise and the covariates (additive, multiplicative and complex) and different noise regimes; $\mathcal{N}$ denotes $\epsilon \sim \mathcal{N}$, $\mathcal{U}$ denotes $\epsilon \sim \mathcal{U}(0,1)$ and Exp denotes $\epsilon \sim \mathrm{Exp}(1)$.

| | Additive Noise | | | Multiplicative Noise | | | Complex Noise | | |
|---|---|---|---|---|---|---|---|---|---|
| (A) | $\mathcal{N}$ | $\mathcal{U}$ | Exp | $\mathcal{N}$ | $\mathcal{U}$ | Exp | $\mathcal{N}$ | $\mathcal{U}$ | Exp |
| LiNGAM | 26% | 87% | 28% | 20% | 30% | 5% | 0% | 2% | 0% |
| ANM | **100%** | **100%** | **100%** | 0% | 0% | 1% | 28% | 26% | 24% |
| PNL | 53% | 14% | 47% | 52% | 24% | 30% | 55% | 50% | 48% |
| IGCI | 52% | 52% | 94% | **100%** | 89% | **100%** | **100%** | 85% | **100%** |
| KCDC | **100%** | **100%** | **100%** | **100%** | **100%** | **100%** | 98% | **92%** | **100%** |
| | | | | | | | | | |
| (B) | $\mathcal{N}$ | $\mathcal{U}$ | Exp | $\mathcal{N}$ | $\mathcal{U}$ | Exp | $\mathcal{N}$ | $\mathcal{U}$ | Exp |
| LiNGAM | 4% | 40% | 4% | 10% | 22% | 4% | 31% | 32% | 23% |
| ANM | 94% | 97% | 79% | 8% | 30% | 12% | 16% | 54% | 6% |
| PNL | 54% | 33% | 46% | 49% | 58% | 32% | 56% | 50% | 72% |
| IGCI | 54% | 68% | 96% | **100%** | 89% | **100%** | 88% | 72% | 97% |
| KCDC | **100%** | **100%** | **100%** | **100%** | **100%** | **100%** | **100%** | **100%** | **100%** |
| | | | | | | | | | |
| (C) | $\mathcal{N}$ | $\mathcal{U}$ | Exp | $\mathcal{N}$ | $\mathcal{U}$ | Exp | $\mathcal{N}$ | $\mathcal{U}$ | Exp |
| LiNGAM | 25% | 32% | 18% | 0% | 3% | 0% | 0% | 0% | 1% |
| ANM | 98% | **100%** | 97% | 5% | 1% | 0% | 31% | 19% | 37% |
| PNL | 39% | 27% | 36% | 55% | 41% | 30% | 95% | 92% | 92% |
| IGCI | 98% | **100%** | 99% | **100%** | 99% | **100%** | 97% | 98% | 98% |
| KCDC | **100%** | **100%** | **100%** | **100%** | **100%** | **100%** | **100%** | **100%** | **100%** |