[Reviews · NeurIPS 2018]

Reviewer 1



This paper proposes a novel approach to causal discovery based on kernel conditional deviance methods. I found the paper very well structured and well written. After setting out the problem and giving a nice visual motivation for their approach, the main contributions are clearly described, and the related work is very nicely summarized in the next section. In the description of the proposed approach, the notation was well introduced, the approach is quite intuitive and accessible through the provided description, and I didn’t find any problems with the mathematical description and derivations. Finally, the set-up of the simulation study was well described so that I would expect to be able to reproduce the results, and the results are quite impressive in that the proposed approach performs well in comparison to existing approaches. A few minor comments: - For me, the motivating example in the introduction would have been easier accessible with more details which values of x and y, respectively were used in generating the plots. - In the simulation experiments in Table 3 in the supplements, it is nice that their proposed method achieves perfect performance in almost all scenarios, but that makes me wonder when it does not achieve perfect performance and how it compares in these scenarios to the alternative methods. I.e. I would be interested in more simulations with larger noise. Furthermore, I would be interested in seeing how KCDC performs for linear or quadratic association between X and Y. - There were a couple of minor typos which could be removed from one more proofread.

Reviewer 2



In this manuscript, the authors propose a fully nonparametric method called KCDC to discover causal relationships from observational data. The method is motivated by the observation that there should be an asymmetry between cause and effect in terms of the variability. An applicable measure using the framework of RKHS space is derived. The good performance is verified in both synthetic and real-world datasets. Strengths: 1. Overall, I didn’t find any obvious weakness or error in the paper. I think the contribution is solid, and the writing and logic are quite clear. 2. The framework does not postulate a priori assumption on the functional structure, overcoming some problems in the literature. 3. Using the RKHS framework makes the approach widely applicable in many data situations. Minor comments: 1. I think the most relevant literature is [11]. It’s better to compare with the work more in the method part, or compare with the work in more details. 2. As the most relevant work, it’s better to put [11] in the benchmarks. 3. The detailed description of Figure 1 is lacking, making it confused at first view.

Reviewer 3



This paper provides a tractable method for identifying causality in bivariate data. In particular, they posit that the Kolmogorov complexity of the effect's generating mechanism is independent of the value of cause, whereas the opposite does not hold. The authors appeal to heuristics and do not provide any theory regarding identifiability, but the method works very well across a wide variety of simulation settings and on a standard benchmark data set. Comments: 1) The authors state that they variety of kernels for the input and response "yielded fairly consistent performance" and they report the results for one specific setting. Is there an intuition as to which kernels would work better in different situations? Or should you be using the same kernel for cause and effect? In the paper, they use 2 different kernels. 2) Roughly what is the computational cost of determining a single causal relationship? The authors mention in section 5 that this could be extended to multivariate data, but would taht be computationally feasible? 3) It might be interesting to also look at the "level" of the test. For instance, in the case where we know the graph is unidentifiable (linear gaussian for example), what does the distribution of T^{KCDC} look like?